# SARS-CoV-2 proteome microarray for global profiling of COVID-19 specific IgG and IgM responses

He-wei Jiang [1,6], Yang Li[1,6], Hai-nan Zhang[1,6], Wei Wang[2,6], Xiao Yang[4], Huan Qi[1], Hua Li[5], Dong Men [3✉], Jie Zhou[2✉] & Sheng-ce Tao [1✉]

We still know very little about how the human immune system responds to SARS-CoV-2. Here we construct a SARS-CoV-2 proteome microarray containing 18 out of the 28 predicted proteins and apply it to the characterization of the IgG and IgM antibodies responses in the sera from 29 convalescent patients. We find that all these patients had IgG and IgM antibodies that specifically bind SARS-CoV-2 proteins, particularly the N protein and S1 protein. Besides these proteins, significant antibody responses to ORF9b and NSP5 are also identified. We show that the S1 specific IgG signal positively correlates with age and the level of lactate dehydrogenase (LDH) and negatively correlates with lymphocyte percentage. Overall, this study presents a systemic view of the SARS-CoV-2 specific IgG and IgM responses and provides insights to aid the development of effective diagnostic, therapeutic and vaccination strategies.

[1] Shanghai Center for Systems Biomedicine, Key Laboratory of Systems Biomedicine (Ministry of Education), Shanghai Jiao Tong University, 200240 Shanghai, China. [2] Foshan Fourth People's Hospital, 528000 Foshan, China. [3] State Key Laboratory of Virology, Wuhan Institute of Virology, Chinese Academy of Sciences, 430071 Wuhan, China. [4] Key Laboratory of RNA Biology, Institute of Biophysics, Chinese Academy of Sciences, 100101 Beijing, China. [5] Bio-ID Center, School of Biomedical Engineering, Shanghai Jiao Tong University, 200240 Shanghai, China. [6] These authors contributed equally: He-wei Jiang, Yang Li, Hai-nan Zhang, Wei Wang. ✉email: d.men@wh.iov.cn; zjet65@163.com; taosc@sjtu.edu.cn

COVID-19 is caused by the coronavirus SARS-CoV-2[1,2]. It is presently recognized by the World Health Organization as a global pandemic, and as of June 28, 2020, 9,653,066 diagnosed cases have been reported from 214 countries, areas or territories (http://2019ncov.chinacdc.cn/2019-nCoV/). Sequence analysis suggested that SARS-CoV-2 is most closely related to the BatCoV RaTG13 and belongs to the subgenus, sarbecovirus, of the beta coronaviruses, together with the Bat-SARS-like coronavirus and the SARS coronavirus[1,2]. By comparing SARS-CoV to the other related coronaviruses, it was predicted that there are 28 proteins encoded in the genome of SARS-CoV-2[3]. Further, such comparisons suggested that SARS-CoV-2 might utilize the same mechanism to enter the host cells, namely via high-affinity binding between the receptor-binding domain (RBD) of the spike protein (S protein) and angiotensin converting enzyme 2 (ACE2)[4–9].

Though there is presently tremendous worldwide effort to identify and develop effective therapeutic approaches against this virus, none of this work has been successful at the moment. One possible approach that has shown some positive results is by treating infected patients with the plasma collected from convalescent COVID-19 patients[10,11]. Here, it is believed that the humoral antibody response in these convalescent patients played an important role in their recovery, and so might likewise prove effective in other, presently infected patients. Indeed, it is known that in combating many viral infections, including SARS-CoV and MERS-CoV, IgG, and IgM antibodies play many critical roles[12–15].

However, because SARS-CoV-2 is a newly emerged pathogen, the precise IgG and IgM responses in the COVID-19 patients are very poorly understood. Indeed, in this regard, there are many important questions that need to be experimentally addressed: (1) What is the variation among different patients, especially for antibodies against the nucleocapsid protein (N protein) and S protein? (2) Are there any other viral proteins that could trigger significant antibody responses in at least some of the patients? (3) Is it possible to link the magnitude of the overall IgG and IgM response to the severity of the disease in patients? Resolution of these questions is fundamental to the development of an understanding of the global IgG and IgM responses against SARS-CoV-2 and for the possibility to use this material in the development of effective therapeutic or diagnostic approaches.

Conventional techniques for studying patient IgG and IgM responses include ELISA[16–18] and the immune-colloidal gold strip assay[17,19,20]. However, these techniques usually can only test a single target protein or antibody in a single reaction. By contrast, protein microarrays enable proteome-wide characterization of antibody responses in a high-throughput format, providing a more systemic description of these vital antibody responses. Indeed, a variety of protein microarrays have already been constructed and successfully applied to serum antibody profiling, such as the *Mycobacterium tuberculosis* proteome microarray[21], the SARS-CoV protein microarray[12], the Dengue virus protein microarray[22] and the influenza virus protein microarray[23].

Here, we describe the construction of the SARS-CoV-2 proteome microarray and its application in the characterization of the global IgG and IgM responses from 29 COVID-19 convalescent patients. In this way, we provide a systemic view of these responses, revealing both common and unique features of these patients, which may aid future diagnostic and therapeutic efforts against this virus.

## Results

**Schematic diagram and workflow**. The genome of SARS-CoV-2 is ~29.8 kb and is predicted to encode for 28 proteins[3]:

5 structural proteins (treating the S protein as two separate proteins, S1 and S2), 8 accessory proteins, and 15 non-structural proteins (nsp) (Fig. 1a). The corresponding nucleotide sequences of all of these proteins and the receptor-binding domain (RBD) of the S1 protein were synthesized and cloned into appropriate vectors for expression in *E. coli*, and the expressed proteins were purified by affinity chromatography. To obtain any even broader range of proteins that were produced from different prokaryotic and eukaryotic systems, we also acquired a number of recombinant SARS-CoV-2 proteins from commercial sources (Supplementary Data 1). After evaluating the proteins for quality control, these proteins were printed on appropriate substrate slides. Convalescent sera were collected from 29 patients on the day of their discharge and were applied to the proteome microarray. We detected the SARS-CoV-2-specific IgG and IgM proteins bound to the array using fluorescent-labeled anti-human antibodies, thereby generating a global assessment of each patient's humoral antibody response.

**Generation of the predicted SARS-CoV-2 proteins**. To produce the recombinant proteins of SARS-CoV-2 for the microarray, we first determined the amino-acid sequences of the predicted proteins[3] based on the reference genome (Genbank accession No. MN908947.3). We split S protein into S1 and S2, as suggested previously[3], to enable a more precise analysis and also included the RBD alone owing to its critical role during the entry of SARS-CoV-2 into the cells. The protein sequences were converted to the corresponding nucleotide sequences, followed by optimization of the sequences, and then insertion of the sequences into expression vectors (pET32a or pGEX-4T-1). The final expression library included 31 clones. Further information about these clones is included in Supplementary Data 1. After several rounds of optimization, we successfully purified 17 of these proteins (Supplementary Fig. 1). Western blotting with an anti-6xHis antibody and Coomassie staining showed that most of the SARS-CoV-2 proteins exhibit clear bands with the expected size (±10 kDa) and good purity.

To cover the proteome of SARS-CoV-2 as complete as possible, and to include proteins with post-translational modifications (PTM), especially glycosylation, we also acquired recombinant SARS-CoV-2 proteins produced using yeast cell-free systems or mammalian cell expression systems from a variety of commercial sources (Supplementary Fig. 1). Among the collected proteins, there are several different full length and fragmented versions of the S and N proteins (Supplementary Data 1). In this way, we finally obtained 37 proteins from different sources, covering 18 out of the 28 predicted proteins of SARS-CoV-2, that were of suitable concentration and purity for microarray construction.

**Fabrication of the SARS-CoV-2 protein microarray**. A total of 37 proteins along with positive and negative controls were printed on the microarray slide (Fig. 2a). Since most of the proteins were tagged with the 6xHis peptide, we examined the overall quality of the microarray by probing with an anti-6xHis antibody, which revealed uniform, spot-limited labeling across the entire microarray, thus attesting to the quality of the array (Fig. 2a). In addition, we noticed that NSP7 was contaminated during the microarray manufacturing process. Thus, we decided not to include NSP7 for further analysis.

When probed with convalescent sera from COVID-19 patients, we generally observed high, multi-spot antibody responses, which were not observed with the control sera (Fig. 2b). To prevent or largely decrease nonspecific signals generated from the background of the expression system and minimize any influence from possible protein impurity, *E. coli* lysates and eGFP were

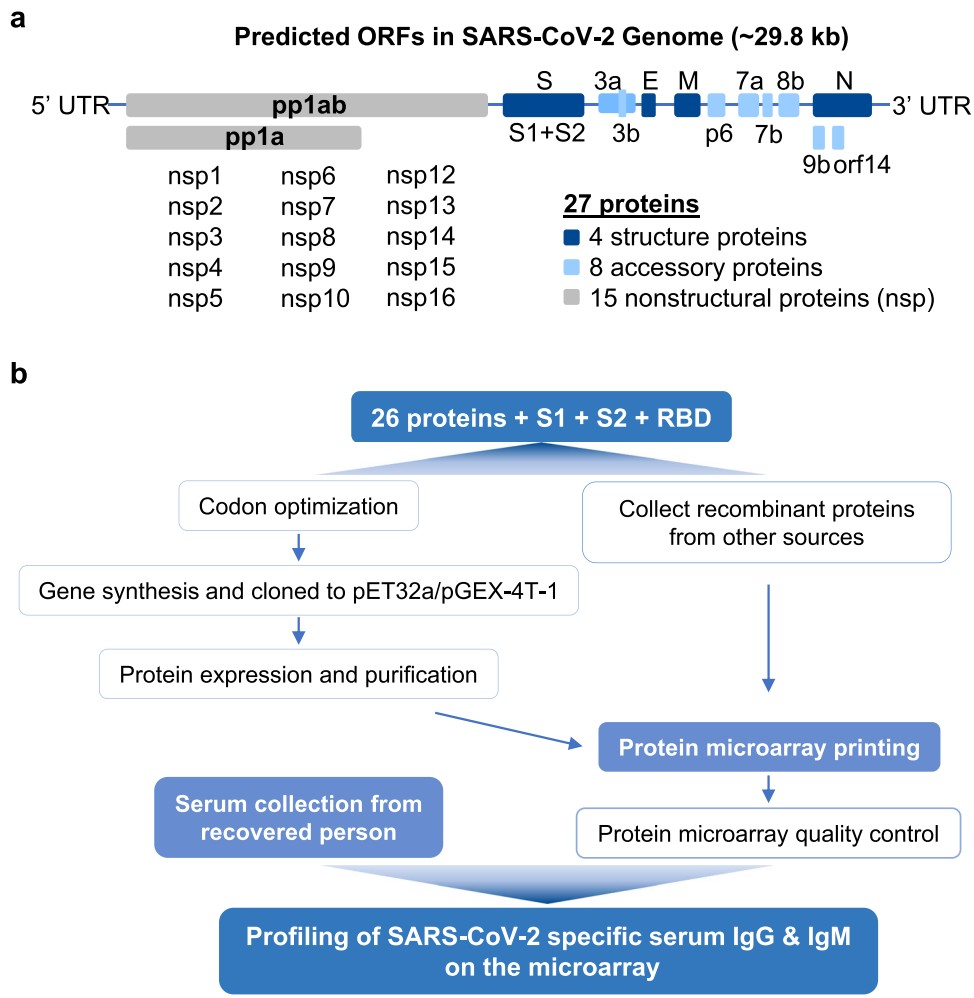

**Fig. 1 The workflow of SARS-CoV-2 proteome microarray fabrication and serum profiling. a** The genome of SARS-CoV-2 and the 28 predicted proteins. **b** The workflow of proteome microarray fabrication and serum profiling on the microarray.

added during the incubation with serum samples, which significantly reduced nonspecific signals (Supplementary Fig. 2a). To test the experimental reproducibility of the serum profiling using the microarray, we randomly selected two COVID-19 convalescent sera and probed them on three separate microarrays. The Pearson correlation coefficients from the measured intensities over the entire array between two samples were 0.988 and 0.981 for IgG and IgM, respectively. Further, the overall fluorescence intensity ranges of the repeated experiments were quite similar, demonstrating a high reproducibility of the microarray-based serum profiling both for IgG and IgM (Fig. 2c–e).

**SARS-CoV-2-specific serum antibody profiles revealed by proteome microarray**. To globally profile the antibody response against the SARS-CoV-2 proteins from the serum of COVID-19 patients, we screened sera from 29 convalescent patients, along with 21 controls, using the SARS-CoV-2 proteome microarray. The patients were hospitalized in Foshan Fourth hospital in China from 2020-1-25 to 2020-2-27 for various durations. Patient information is summarized in Table 1. Serum from each patient was collected on the day of hospital discharge when standard criteria were met. All of the samples and the controls were probed on the proteome microarray, and after data filtering and normalization, we constructed the IgG and IgM profile for each serum and performed clustering analysis to generate heatmaps

(Figs. 3–4 and Supplementary Figs. 3–4). The patients and controls formed clearly separate clusters for both IgG and IgM data. As expected, the N and S1 proteins elicited high antibody responses in almost all patients but were associated with only weak signals in control groups, confirming the efficacy of these two proteins for diagnosis. Interestingly, we also found that in some cases, proteins such as ORF9b or NSP5 can generate significantly high signals compared with that in the control groups. To further prove the specificity, we performed an immunoblotting-based serum analysis. As expected, the serum specifically recognized ORF9b, S proteins and N proteins (Supplementary Fig. 2b).

**Strong antibody responses against S and N proteins**. Since S and N proteins have been widely used as antigens for diagnosis of COVID-19, we next characterized the serum antibody responses against these two proteins in more detail. With the present cohort, the signals from both the N and S proteins, except for the S1-4 fragment, exhibited strong discriminatory ability between the COVID-19 patients and controls using either IgG or IgM response (Fig. 5a, b, Supplementary Fig. 6a, b). Notably, two sera from the control group exhibited a significant IgG antibody response to the N proteins, with one to N-Nter and the other to N-Cter (Fig. 5g), suggesting that the N protein might generate a higher false-positive measurement than the S protein, especially the S1 protein. To investigate the consistency of signal intensities

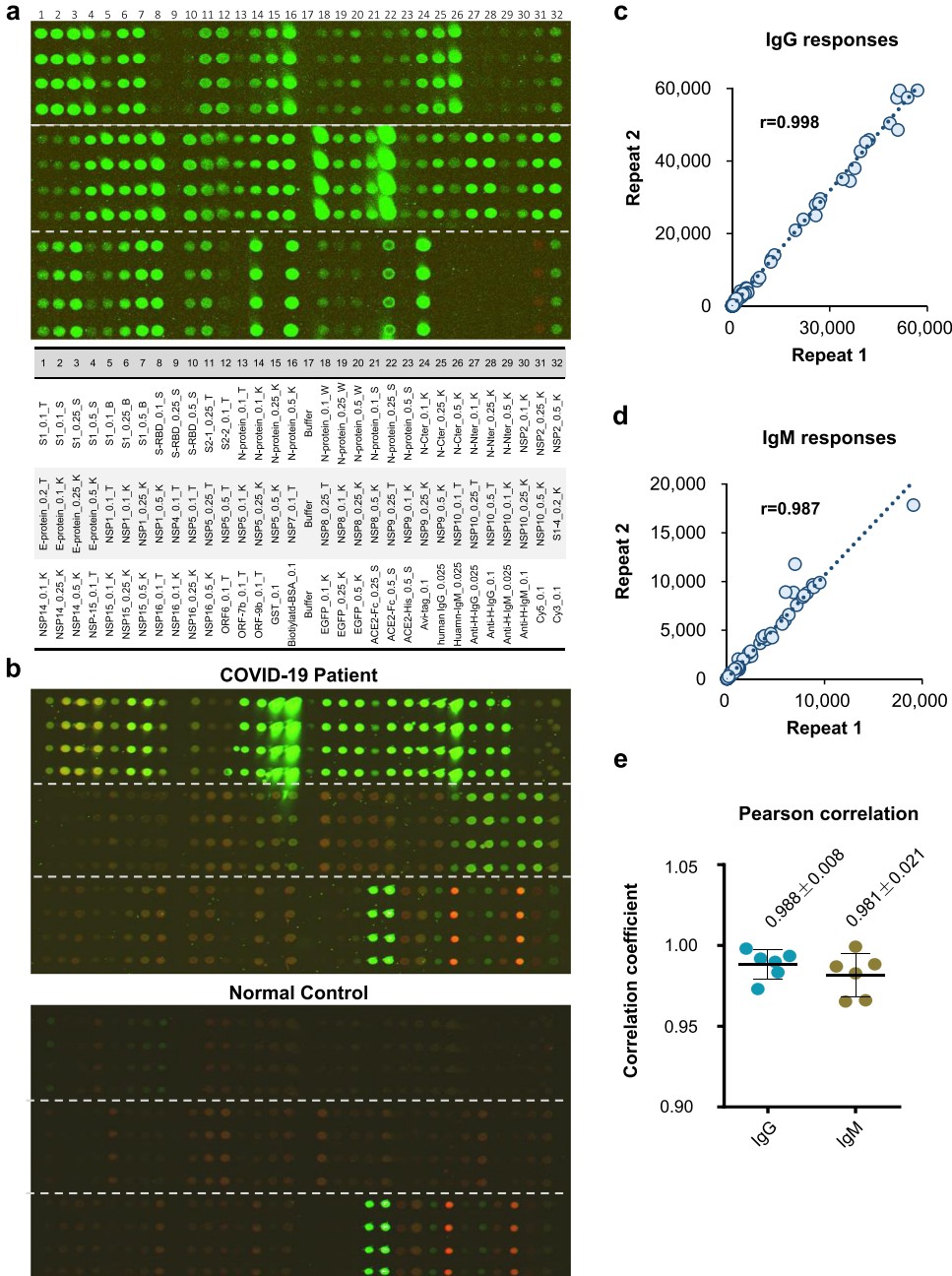

**Fig. 2 SARS-CoV-2 proteome microarray layout and quality control. a** There are 14 identical subarrays on a single microarray. A microarray was incubated with an anti-6xHis antibody to demonstrate the overall microarray quality (green). One subarray was shown. The proteins were printed in quadruplicate. The triangles indicate dilution titers of the same proteins. **b** Representative subarrays probed with sera of a COVID-19 convalescent and healthy control. The IgG and IgM responses were shown in green and red, respectively. **c, d** The correlations of the overall IgG and IgM signal intensities between two repeats probed with the same serum. Proteins ($n = 93$) on the microarray were examined. **e** Statistics of the Pearson correlation confidence among repeats probed with the same serum. Two serum samples from the convalescent group was examined in three independent experiments. NC negative control, PC positive control; 0.1, 0.2, 0.25, and 0.5 indicate the concentration of these proteins for microarray printing. T Tao Lab, B Hangzhou Bioeast biotech. Co.,Ltd., K Healthcode Co., Ltd., S Sanyou biopharmaceuticals Co.,Ltd., W VACURE l Biotechnology Co.,Ltd., Y Sino biological Co.,Ltd. Expression system: (1) *E. coli*: All proteins from Tao Lab (T), N Protein _S, N Protein_W; (2) Cell-free: All proteins from Healthcode Co., Ltd. (K), (3) Mammalian: S1_B, S1_S, S-RBD_S, S-RBD_Y.

among the different sources, full length or fragments of these proteins, we calculated the Pearson correlation coefficients among the S proteins (Fig. 5c, Supplementary Fig. 6c) and N proteins (Supplementary Fig. 5f) using data of the convalescent sera. High correlations were observed among different concentrations of the same proteins as well as the same protein from different sources

(Fig. 5d, h, Supplementary Figs. 5a–c, g and 6d, g), although the N protein at high concentrations generated almost saturated IgG signals (Supplementary Fig. 5g). In particular, for the full-length S1 proteins from different sources, whether from *E. coli* (S1_T) or 293 T (S1_B and S1_S) expression systems, a high correlation between these proteins were observed (Fig. 5c–d, Supplementary

Fig. 6c, d), indicating that the S1 proteins from different sources that we have tested are all similarly effective for detection. However, the background signals in the control group were much lower for proteins purified from mammalian cells (such as 293 T) (Fig. 5a, Supplementary Fig. 6a), suggesting that these samples might possess a higher specificity and could serve as better reagents for developing immune diagnostics. The signals of the full-length S1 protein were highly correlated with that of the S-

RBD (Fig. 5e, Supplementary Fig. 6e) but with much stronger signals. In contrast, the correlation levels of the S1-4 fragment with the full-length S1 or RBD were lower (Fig. 5c, Supplementary Fig. 5d). Also, the S1 signals were poorly correlated with S2 proteins, although significant S2 signals were observed for many of the patients (Fig. 5f, Supplementary Figs. 5e and 6f). These data might reflect a difference in the immunogenicity of different regions of the S protein, which could be resolved in the future with more refined epitope mapping. Similar results were also observed for the N proteins (Supplementary Fig. 5h, i). Interestingly, a moderate but significant linear correlation was observed between the IgG responses against the N and S1 proteins (Fig. 5i) but not the IgM responses (Supplementary Fig. 6h), while the correlations between the IgG and IgM signals for the same protein were low (Fig. 5j, k). This might partially be a consequence of overall lower IgM signals than the IgG signals (Supplementary Fig. 6a, b) at the convalescent stage.

**Antibody responses against other proteins.** To statistically analyze the IgG responses against SARS-CoV-2 proteins, we calculated the $p$-values followed by multiple testing correction (or $q$-values), and applied significant analysis of microarray (SAM) to identify significant positive proteins (Supplementary Fig. 7 and Data 2). Besides S and N proteins, ORF9b and NSP5 also had significant positive responses. Particularly, 44.8% (13/29) and 10.3% (3/29) patients exhibited a "positive" IgG antibody signal to ORF9b and NSP5, respectively (Fig. 6a–c). Although E protein and ORF7b were statistically positive, the fluorescent intensities in both patient and control groups were too low, further verification is needed for these two proteins. To investigate if the IgG responses against ORF9b or NSP5 depended on the IgG responses to the N or S proteins, we calculated the correlations between these measurements. We observed no obvious correlation between the IgG signals to ORF9b or NSP5 and the IgG

| Table 1 Detailed information on serum samples tested in this study. | | |
|---|---|---|
| **Patient group** | | $n = 29$ |
| Gender | Male | 13 |
| | Female | 16 |
| Age (years) | | 42.3 ± 13.8 |
| Severity | Mild cases | 3 |
| | Moderate cases | 26 |
| Days after onset | | 22.3 ± 5.4 |
| Hospital stay (days) | | 17.9 ± 5.7 |
| Sample collection data | | 20200125–20200227 |
| **Control group** | | $n = 21$ |
| Lung cancer patients | | 10 |
| Gender | Male | 5 |
| | Female | 5 |
| Age (years) | | 55.9 ± 7.3 |
| Sample collection data (year) | | 2017 |
| Health control | | 11 |
| Gender | Male | 6 |
| | Female | 5 |
| Age (years) | | 45.1 ± 12.9 |
| Sample collection data (year) | | 2017–2018 |

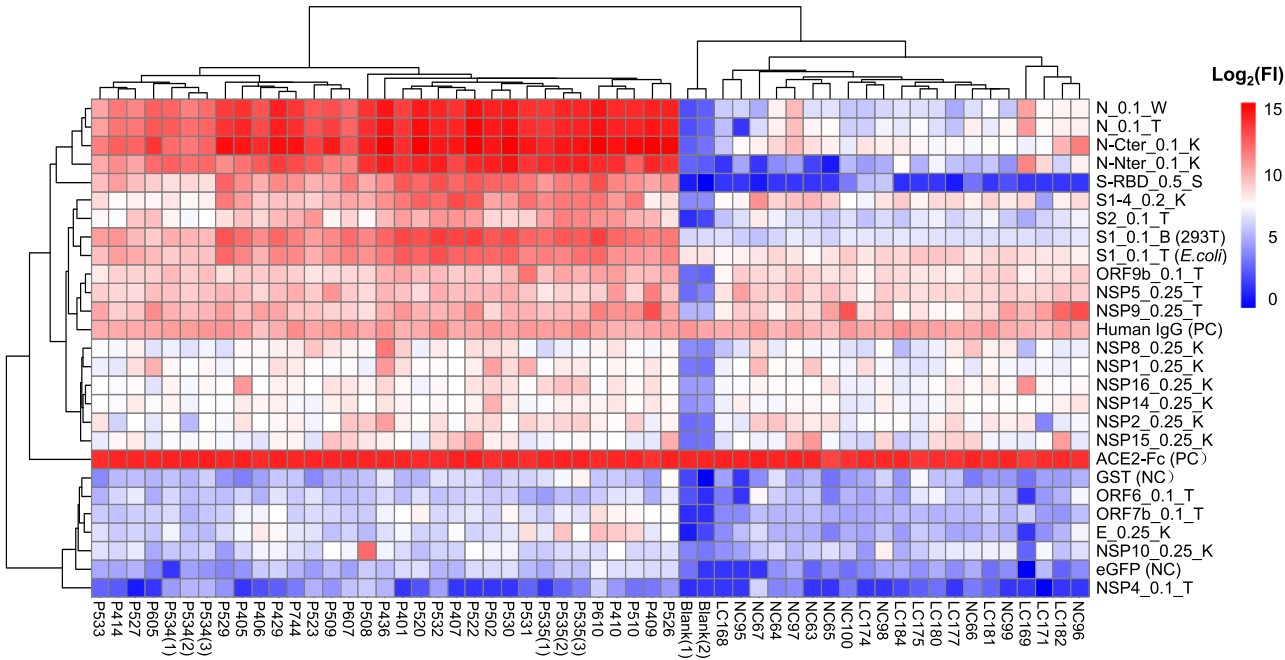

**Fig. 3 The overall SARS-CoV-2-specific IgG profiles of the 29 convalescent sera against the proteins.** Each square indicates the IgG antibody response against the protein (row) in the serum (column). Proteins were shown with names along with concentrations (µg mL$^{-1}$) and sources. Sera were shown with group information and serum number. NCP Novel Coronavirus Patients or COVID-19 patients, LC lung cancer, NC normal control. Blank means no serum. Three repeats were performed for serum P534 and P535. FI fluorescence intensity.

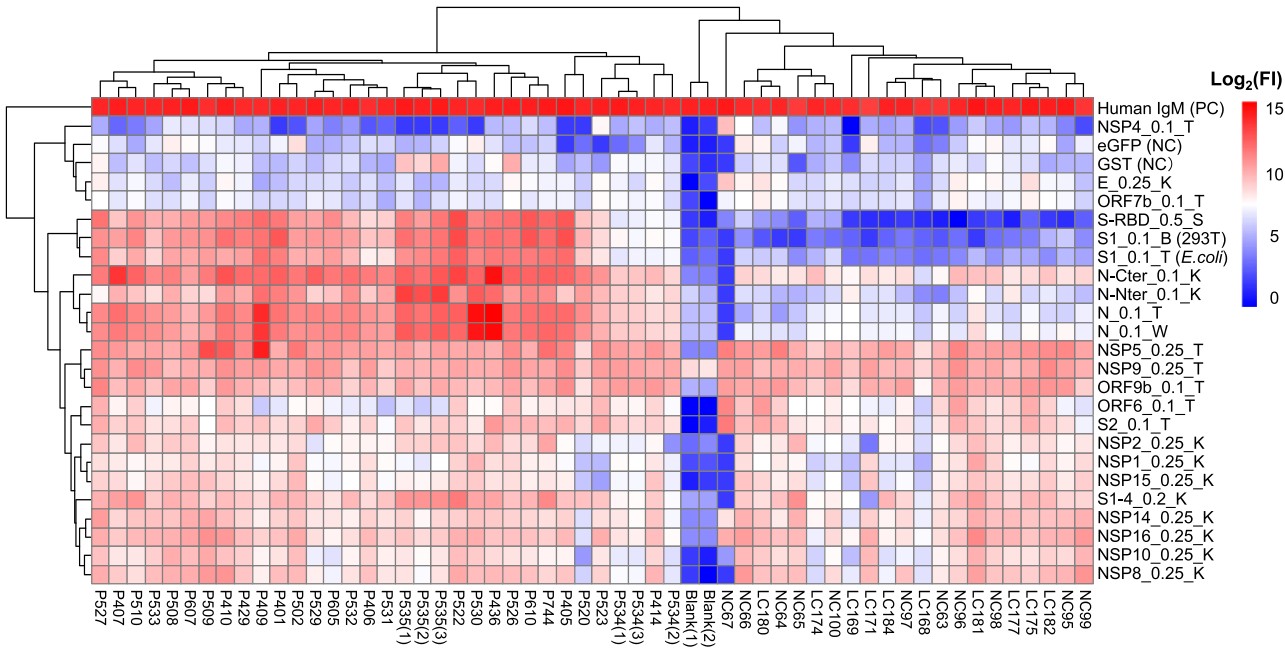

**Fig. 4 The overall SARS-CoV-2-specific IgM profiles of the 29 convalescent sera against the proteins.** Each square indicates the IgM antibody response against the protein (row) in the serum (column). The rest was the same as that of Fig. 3.

signals to the N or S proteins (Fig. 6d, e), suggesting these two proteins might provide complementary information to that generated from the N or S proteins, either for diagnosis or efforts to understand the specific immune response to this virus.

**IgG responses were correlated with age, LDH, and lymphocyte percentage.** It is known that the immune response is closely related to the development of the disease in individual patients. To study the relationship between the antibody response and the course of the disease, we examined the correlations between the S1 IgG responses to various proteins with clinical characteristics. Not surprisingly, the time after disease onset correlated with the IgG response against the S1 (Fig. 7a) as the IgG response usually increases over time and reaches a maximum several weeks after disease onset, as observed in other studies[24] and SARS patients[25]. We also found that age also correlated with the IgG response to the S1 (Fig. 7b).

We also found that the IgG responses against S1 protein were positively correlated with peak lactate dehydrogenase (LDH) levels and inversely correlated with percentage of lymphocyte (Ly %) (Fig. 7c, d). It was also demonstrated that the IgG responses were slightly different between male and female patients (Fig. 7e).

We further performed multiple linear regression to investigate the relationship among S1 IgG level, age, gender, days after onset, peak LDH and Ly% (Fig. 7f). Consistent with above correlation analysis, age and peak LDH were statistically significant (both with p-values <0.05) and gender showed marginally significance ($p = 0.059$). As expected, days after onset, identified as a confounding factor, showed no statistically significant difference ($p = 0.514$) and was removed from the regression. Ly% was still kept in the model as its low significant ($p = 0.106$) was probably due to the small sample size. The final equation (adjusted R-squared = 0.60, p-value < 0.001) is as follows: $y = 1318 + 6155*x_1 + 2166*x_2 - 4548*x_3 + 6842*x_4$, where $y$ represents S1 IgG level and $x_1$, $x_2$, $x_3$, $x_4$ represents the normalized values (between 0 and 1) of age, gender, Ly%, and peak LDH, respectively.

## Discussion

To profile the SARS-CoV-2-specific IgG and IgM responses, we have constructed a SARS-CoV-2 proteome microarray with 18 of the 28 predicted proteins. A set of 29 convalescent sera were analyzed on the microarray, global IgG and IgM profile were obtained simultaneously through a dual color strategy. Our data clearly showed that both N protein and S1 were suitable for diagnostics, while S1 purified from the mammalian cell might possess better specificity. When we were preparing this work, a preprint also found better specificity with mammalian versus insect cell expressed proteins[26]. Meanwhile, significant antibody responses were identified for ORF9b and NSP5. We further showed that the level of S1 IgG positively correlated to age and the level of LDH while negatively correlated to Ly%.

It is well known that S1 and N proteins are the dominant antigens of SARS-CoV and SARS-CoV-2 that elicit both IgG and IgM antibodies, and antibody response against N protein is usually stronger. However, we found for two of the control sera, strong IgG bindings were observed for N protein, and specifically, one control recognizing N protein at the N-terminal while the other at the C-terminal. This may be due to the high conserved N protein sequences across the coronavirus species. This indicating we should be aware of the false-positive when applying N protein for diagnosis. In contrast, S1 protein demonstrated a higher specificity. Thus, an ideal choice of developing immuno-diagnostics might be the combining of both N protein and S1.

We also compared the antibody responses against a variety version of S1, including the full length, the RBD domain, the N-terminal, and the C-terminal. The antibody response to the RBD region was highly correlated with that to full-length protein but with weaker signals which is consistent with a recent study[26], however, the correlations among other S1 versions were not significant, suggesting dominant epitopes that elicit antibodies might differ among individuals. Further study of detailed epitope mapping might give us a clear answer.

In this study, we also found the significant presence of IgG and IgM against ORF9b (13 out of 29 cases) and NSP5 (3 out of 29 cases). ORF9b is predicted as an accessory protein, exhibiting

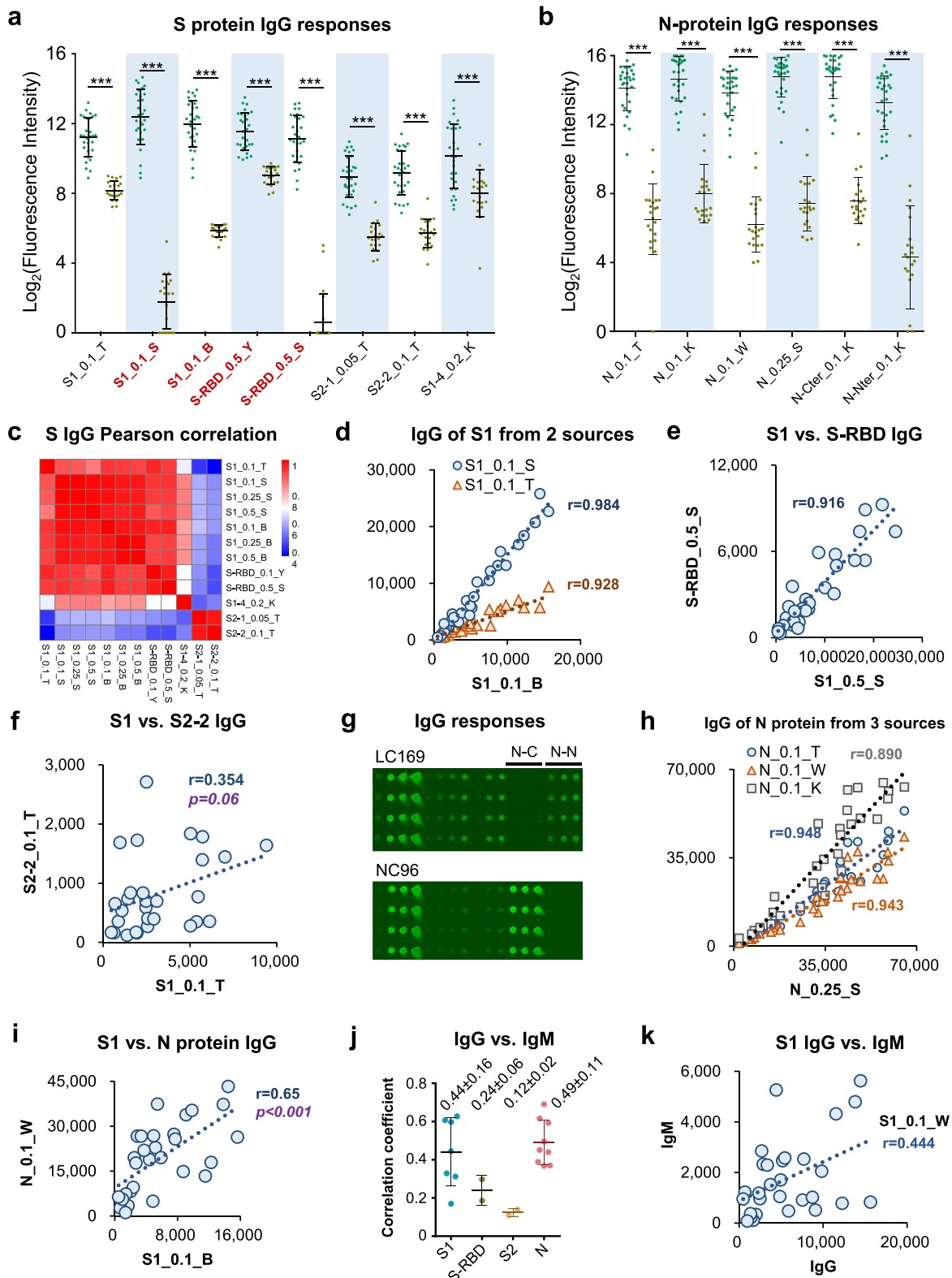

high overall sequence similarity to SARS and SARS-like COVs ORF9b (V23I)[3], and is likely to be a lipid-binding protein[27]. Previous studies showed that SARS ORF9b suppressed innate immunity by targeting mitochondria[28]. Two previous studies have found antibodies against SARS ORF9b presented in the sera of patients recovering from SARS[29,30]. Our study also demonstrates the potential of antibodies against ORF9b for the detection

of convalescent COVID-19 patients. COVID-19 NSP5 is also highly homologous to SARS NSP5 (96% identity, 98% similarity). Its homologous proteins in a variety of coronaviruses have been proven to impair IFN response[31–33]. Our study provide experimental evidence to show the existence of NSP5-specific antibodies in convalescents. Since NSP5 is a non-structural protein, theoretically, it should present only in the infected cells but not in

**Fig. 5 IgG responses to S and N proteins. a** Box plots of IgG response for S1 and S2 proteins. The proteins labeled with bold and red were overexpression in mammalian cell lines. **b** Box plots of IgG response for N proteins. For **a**, **b**, each dot indicates one serum sample either from the convalescent group (green, n = 29) or the control group (brown, n = 21). Data are represented as boxplots where the middle line is the mean value. The upper and lower hinges are mean values ± SD. P values were calculated by the two-sided t-test. Q values were adjusted p-values using BH method. ***q < 0.001. The exact p-values were shown in Supplementary Data 2. **c** Pearson correlation coefficient matrix of IgG responses among different S1 and S2 proteins. **d**–**f** Correlations of overall IgG responses among different S1 proteins (**d**), S1 vs. RBD (**e**) and S1 vs. S2 (**f**). **g** One part of a sub-microarray showed the IgG responses of two controls, i.e., LC169 and NC96 against N proteins, N-Cter and N-Nter indicates the C-terminal and N-terminal of N protein, respectively. **h**, **i** Correlations of the overall IgG responses among different N proteins (**h**) and N protein vs. S protein (**i**). **j** Statistics of the Pearson correlation coefficients between IgG and IgM profile against constructs of S1 (n = 7), S-RBD (n = 2), S2 (n = 2), and N (n = 9). Data are presented as mean values ± SD. **k** Correlations between IgG and IgM profile against S1_0.1_W. For **d**–**f**, **h**–**k**, each dot indicates one serum sample from the convalescent group (n = 29). For **f** and **i**, p-values were calculated by the two-sided t-test.

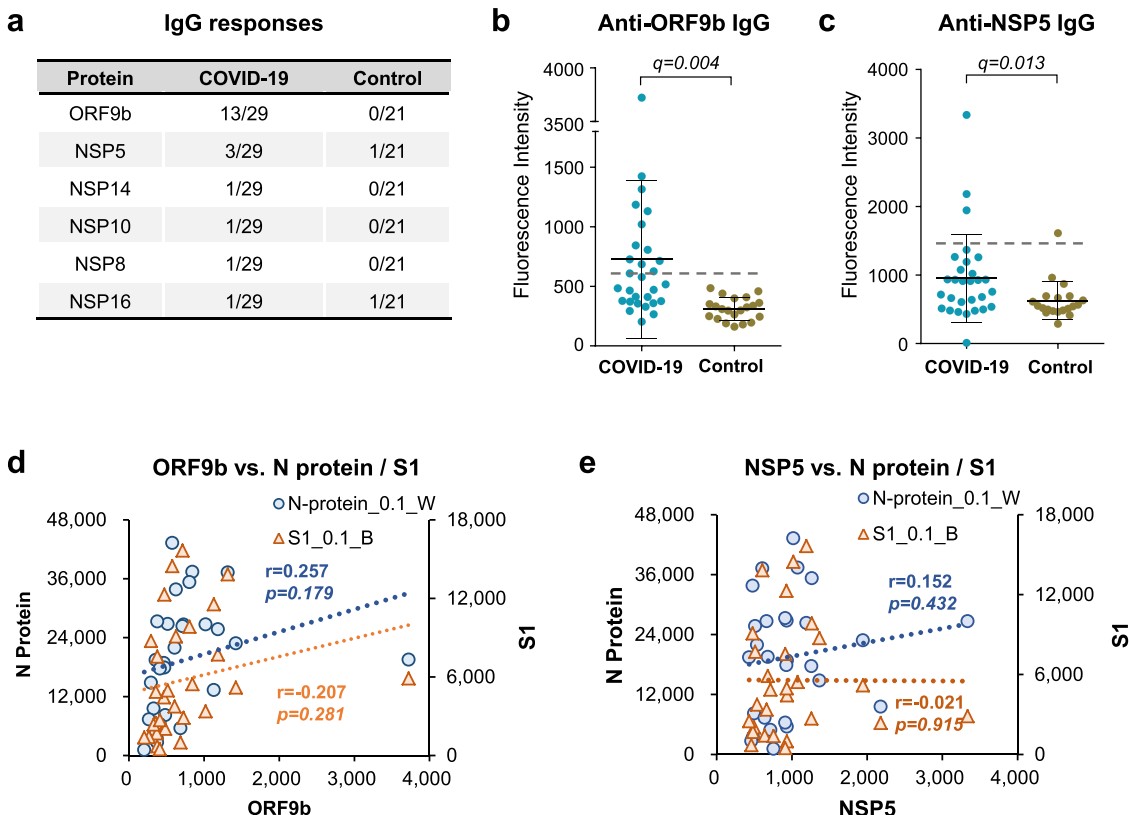

**Fig. 6 IgG response to other SARS-CoV-2 proteins. a** Other SARS-CoV-2 proteins that were recognized by IgG from the convalescent sera, in comparison to that of the controls. **b**, **c** Anti-ORF9b IgG (**b**) or anti-NSP5 IgG (**c**) in the patient and control group. For **b**, **c**, each dot indicates one serum sample either from the convalescent group (n = 29) or the control group (n = 21). Data are presented as mean values ± SD. The dashed line indicates cutoff value calculated as mean + 3x SD of the control group. P-values were calculated by the two-sided t-test and q-values were adjusted p-values using BH method. **d**, **e** Correlations of the overall IgG responses for N or S1 protein vs. ORF9b (**d**) or NSP5 (**e**). For **d**, **e**, each dot indicates one serum sample from the convalescent group (n = 29) and p-values were calculated by the two-sided t-test.

virions. Hence, antibody against NSP5 has the potential to be applied to distinguish between COVID-19 patients and healthy people immunized with the inactivated virus.

We have analyzed the correlations between the COVID-19 specific IgG responses with clinical characteristics as well. It is expected that IgG responses improve over time within one or two months after onset[24,34,35] and we indeed have observed a significant correlation between IgG signals with days after onset. We also found peak LDH was highly correlated with IgG response, especially for female patients. As many studies reported, LDH tends to have a higher level in severe COVID-19 patients and could be an indicator of severity[25,36]. In fact, it has been observed in SARS patients that more severe SARS is associated with more

robust serological response[25,37], a similar association was confirmed in COVID-19 patients.

There are some limitations to the current SARS-CoV-2 proteome microarray. Firstly, due to the difficulty of protein expression and purification, there are still 10 proteins missing[3]. We will try to obtain these proteins through vigorous optimization or other sources. An interesting finding is anticipated in the near future for these missing proteins. Secondly, most of the proteins on the microarray are not expressed in mammalian cells, critical post-translational modifications, such as glycosylation is absent. It is known that there are 23 N-glycosylation sites on S protein, which is heavily glycosylated, and the glycosylation may play critical roles in antibody-antigen recognition[5,38]. Only a few

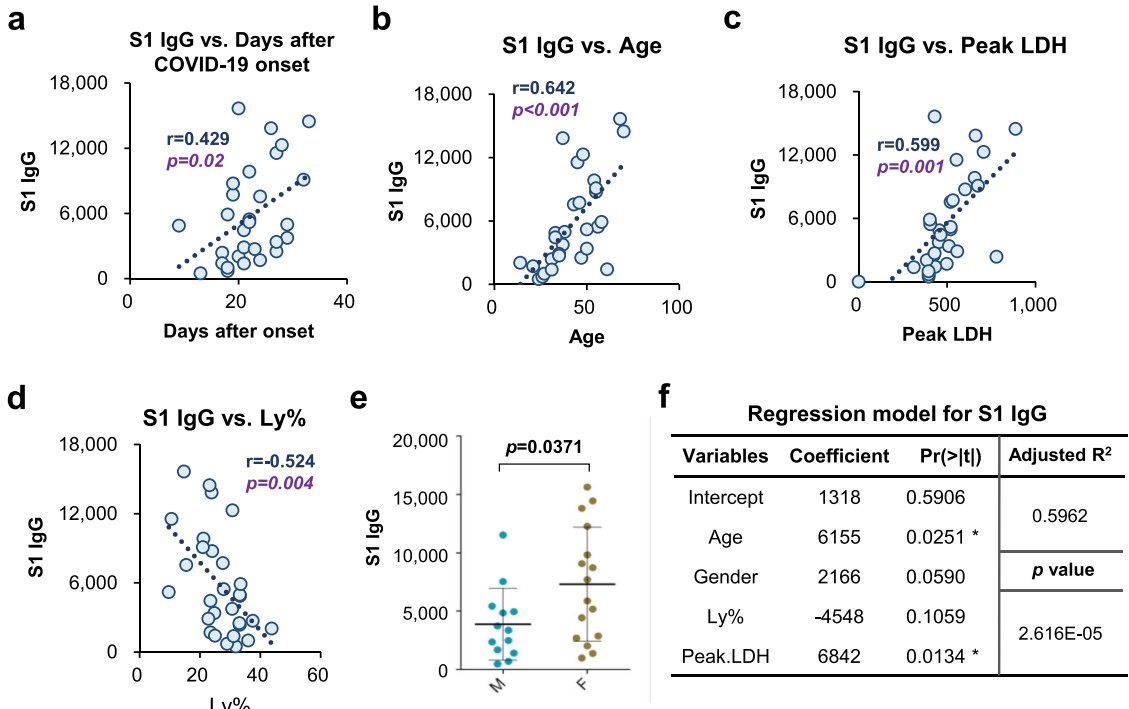

**Fig. 7 Correlations with clinical characteristics. a–d** Correlations of S1 IgG responses with Days after COVID-19 onset (**a**), Age (**b**), peak LDH (**c**), and Ly% (**d**). For **a–d**, each dot indicates one serum sample from the convalescent group ($n = 29$). **e** S1 IgG responses in male (M, $n = 13$) and female (F, $n = 16$) groups. *t*-test Data are presented as mean values ± SD. For **a–e**, *p*-values were calculated by two-sided *t*-test. **f** multiple linear regression model for S1 IgG. *P*-values for coefficient was calculated by two-sided *t*-test and *p*-value for regression model was calculated by one-sided F-test. *$p < 0.05$.

proteins on the current protein microarray were prepared using mammalian cell systems. We are trying the rest of the proteins. Once the microarray is upgraded with many or all proteins purified from mammalian cells, PTM-specific IgG and IgM response may be better elicited. Thirdly, only 29 samples at collected at a single time point were analyzed. Though there are some interesting findings, we believe some of the current conclusions could be strengthened by including more samples. Furthermore, longitudinal samples[39,40] collected at different time points from the same individual after diagnosis or even after cured may enable us to reveal the dynamics of the SARS-CoV-2-specific IgG and IgM responses. The data may be further linked to the severity of COVID-19 among different patients.

The application of the SARS-CoV-2 proteome microarray is not limited to serum profiling. It could also be explored for host-pathogen interaction[41], drug or small molecule target identification[42,43], and antibody specificity assessment[44]. Through the same construction procedure, we could easily expand the microarray to a pan-human coronavirus proteome microarray by including the other two severe coronaviruses, i.e., SARS-CoV[12,45,46] and MERS-CoV[45], as well as the four known mild human coronaviruses[47,48], i.e., CoV 229E, CoV OC43, CoV HKU-1, and CoV NL63. By applying this microarray, we can assess the immune response to coronavirus at a system level, and the possible cross-reactivity could be easily judged.

Taken together, we have constructed the SARS-CoV-2 proteome microarray, this microarray could be applied for a variety of applications, including but not limited to in-depth IgG and IgM response profiling. Through the analysis of convalescent sera on the microarray, we obtained the overall picture of the SARS-CoV-2-specific IgG and IgM profile. We believe that the findings in this study will shed light on the development of the more precise diagnostic kit, more appropriate treatment and effective vaccine for combating the global crisis that we are facing now.

## Methods

**Construction of expression vectors**. The protein sequences of SARS-CoV-2 were downloaded from GenBank (Accession number: MN908947.3). According to the optimized genetic algorithm[49], the amino-acid sequences were converted into *E. coli* codon-optimized gene sequences. Subsequently, the sequences of optimized genes were synthesized by Sangon Biotech. (Shanghai, China). The synthesized genes were cloned into pET32a or pGEX-4T-1 and transformed into *E. coli* BL21 strain to construct the transformants. Detailed information (the DNA sequence, the protein sequence, the size of the protein, the system for protein expression, and etc.) of the clones constructed in this study was given in Supplementary Data 1.

**Protein preparation**. The recombinant proteins were expressed in *E. coli* BL21 by growing cells in 200 mL LB medium to an A600 of 0.6 at 37 °C. Protein expression was induced by the addition of 0.2 mM isopropyl-β-d-thiogalactoside (IPTG) before incubating cells overnight at 16 °C. For the purification of 6xHis-tagged proteins, cell pellets were re-suspended in lysis buffer containing 50 mM Tris-HCl pH 8.0, 500 mM NaCl, 20 mM imidazole (pH 8.0), then lysed by a high-pressure cell cracker (Union-biotech, Shanghai, China). Cell lysates were centrifuged at $12,000 \times g$ for 20 min at 4 °C. Supernatants were purified with Ni$^{2+}$ Sepharose beads (Senhui Microsphere Technology, Suzhou, China), then washed with lysis buffer and eluted with buffer containing 50 mM Tris-HCl pH 8.0, 500 mM NaCl and 300 mM imidazole pH 8.0. For the purification of GST-tagged proteins, cells were harvested and lysed by a high-pressure cell cracker in lysis buffer containing 50 mM Tris-HCl, pH 8.0, 500 mM NaCl, 1 mM DTT. After centrifugation, the supernatant was incubated with GST-Sepharose beads (Senhui Microsphere Technology, Suzhou, china). The target proteins were washed with lysis buffer and eluted with 50 mM Tris-HCl, pH 8.0, 500 mM NaCl, 1 mM DTT, 40 mM glutathione. The purified proteins were analyzed by SDS-PAGE followed by western blotting using an anti-His antibody (Merck Millipore, USA, Cat#05-949) and Coomassie brilliant blue staining. Recombinant SARS-CoV-2 proteins were also collected from commercial sources. Detailed information on the recombinant proteins prepared in this study was given in Supplementary Data 1.

**Protein microarray fabrication**. The proteins, along with the negative (BSA) and positive controls (anti-Human IgG, Cat#I2136 and IgM antibody, Cat#I2386) were printed in quadruplicate on PATH substrate slide (Grace Bio-Labs, Oregon, USA) to generate identical arrays in a $2 \times 7$ subarray format using Super Marathon printer (Arrayjet, UK). Protein microarrays were stored at −80 °C until use.

**Patients and samples**. The Institutional Ethics Review Committee of Foshan Fourth Hospital, Foshan, China approved this study and the written informed consent was obtained from each patient. COVID-19 patients were hospitalized and received treatment in Foshan Forth hospital during the period from 2020-1-25 to 2020-2-27 with variable stay time (Table 1). Serum from each patient was collected on the day of hospital discharge when the standard criteria were met according to Diagnosis and Treatment Protocol for Novel Coronavirus Pneumonia (Trial Version 5), released by the National Health Commission & State Administration of Traditional Chinese Medicine. The basic criteria are the same with that in the Diagnosis and Treatment Protocol for Novel Coronavirus Pneumonia (Trial Version 7)[50]. Briefly, the key points of the discharge criteria are: (1) Body temperature is back to normal for more than three days; (2) Respiratory symptoms improve obviously; (3) Pulmonary imaging shows obvious absorption of inflammation; (4) Nuclei acid tests negative twice consecutively on respiratory tract samples such as sputum and nasopharyngeal swabs (sampling interval being at least 24 h). Sera of the control group from Lung cancer patients and healthy controls were collected from Ruijin Hospital, Shanghai, China. All sera were stored at −80 °C until use.

**Microarray-based serum analysis**. A 14-chamber rubber gasket was mounted onto each slide to create individual chambers for the 14 identical subarrays. The microarray was used for serum profiling as Li, Y. et al.[39] with minor modifications. Briefly, the arrays stored at −80 °C were warmed to room temperature and then incubated in blocking buffer (3% BSA in 1 × PBS buffer with 0.1% Tween 20) for 3 h. Serum samples were diluted 1:200 in PBS containing 0.1% Tween 20, added with 0.1 mg mL$^{-1}$ eGFP purified in the same manner as the eGFP tagged proteins and 0.5 mg mL$^{-1}$ total E. coli lysate. A total of 200 μL of diluted serum or buffer only was incubated with each subarray overnight at 4 °C. The arrays were washed with 1 × PBST and bound antibodies were detected by incubating with Cy3-conjugated goat anti-human IgG and Alexa Fluor 647-conjugated donkey anti-human IgM (Jackson ImmunoResearch, PA, USA, Cat#109-165-008 and Cat#709-605-073 respectively), which were diluted 1: 1000 in 1 × PBST and incubated at room temperature for 1 h. The microarrays were then washed with 1 × PBST and dried by centrifugation at room temperature and scanned by LuxScan 10K-A (CapitalBio Corporation, Beijing, China) with the parameters set as 95% laser power/PMT 550 and 95% laser power/PMT 480 for IgM and IgG, respectively. The fluorescent intensity data was extracted by GenePix Pro 6.0 software (Molecular Devices, CA, USA).

**Immunoblotting-based serum analysis**. The selected proteins were analyzed by SDS-PAGE followed by western blotting using a serum overnight at 4 °C. To assure the quality of the proteins, an anti-His antibody (Merck Millipore, USA, Cat#05-949) was also blotted. The serum for immunoblotting was diluted 1:200 in PBS containing 0.1% Tween 20, with the addition of 0.1 mg mL$^{-1}$ eGFP purified in the same manner as the eGFP tagged proteins and 0.5 mg mL$^{-1}$ total E. coli lysate as mentioned before.

**Statistics**. Signal Intensity was defined as the median of the foreground subtracted by the median of background for each spot and then averaged the quadruplicate spots for each protein. IgG and IgM data were analyzed separately. Before processing, data from some spots, such as NSP7_0.1_T and NSP9_K, were excluded for probably printing contamination. Pearson correlation coefficient between two proteins or indicators and the corresponding p-value was calculated by SPSS software under the default parameters. Cluster analysis was performed by pheatmap package in R[51]. P-values for statistical analysis were calculated by two-way t-test and q-values or adjusted p-values were obtained using BH (Benjamini and Hochberg) method. Significant analysis of microarray (SAM) was performed by "samr" package of the R language with default parameters[52]. To calculate the positive rate of antibody response for each protein, mean signal + 3* standard deviation (SD) of the control sera were used to set the threshold. The multiple linear regression was perfomed with the function "lm" from the "stats" package of the R language. To make the cofficients in the regression model more comparable with each other, the values of all predictor vairables (x) have been normalized as follows: $(x - min(x))/(max(x) - min(x))$.

**Reporting summary**. Further information on research design is available in the Nature Research Reporting Summary linked to this article.

## Data availability

The protein sequences of SARS-CoV-2 were downloaded from GenBank (Accession number: MN908947.3). The SARS-CoV-2 proteome microarray data are deposited in Protein Microarray Database under the accession number PMDE241 (http://www.proteinmicroarray.cn/index.php?option=com_experiment&view=detail&experiment_id=241). Additional data related to this paper may be requested from the authors.

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

## Acknowledgements

We thank Dr. Daniel M. Czajkowsky for English editing and critical comments. We thank Dr. Min Guo of Healthcode Co., Ltd. for providing affinity-purified proteins. We thank Dr. Guo-Jun Lang of Sanyou Biopharmaceuticals Co., Ltd. for providing proteins and antibodies. We also thank Dr. Jie Wang of VACURE l Biotechnology Co., Ltd., Dr. Yin-Lai Li of Hangzhou Bioeast biotech. Co., Ltd., and Sino Biological Co., Ltd. for providing the proteins. This work was partially supported by the National Key Research and Development Program of China Grant (No. 2016YFA0500600), National Natural Science Foundation of China (Nos. 31970130, 31600672, 31670831, 31370813, 31900112, and 21907065) and Foshan Scientific and Technological Key Project for COVID-19 (NO:2020001000430).

## Author contributions

S.-C.T. developed the conceptual ideas and designed the study. H.-W.J., Y.L, H.-N.Z., and H.Q. performed the experiments. J.Z., W.W., D.M., and X.Y. collected the sera samples and provided key reagents. H.L. performed data analysis. S.-C.T., Y.L., and H.-W.J. wrote the manuscript with suggestions from other authors.

## Competing interests

The authors declare no competing interests.
