## [Peer Review File · Nature Communications]

Reviewers' Comments:

Reviewer #1:

Remarks to the Author:

This is a good paper showing that protein microarrays can be applied to rapidly generate outbreak data relevant to the development of diagnostic test, serosurveillance, and vaccine discovery.

Figure 3 has much of the data from this study in a heat map. Most of the antigen features on the array are either non-structural or the N protein or S1. The antigen feature labels are not clear enough and a reviewer can't translate what each one is. The proteins should all be given in a table or spreadsheet with their amino acid coordinates. All of the N protein variants discriminate between COVID cases and non-COVID controls. The non-structural proteins are not immunogenic and they don't recognize antibodies that differentiate cases and controls. It's not clear from the antigen labels what each feature is. The authors conclude that S2 is not reactive, but the heatmap seems to show that it is differentially reactive. The data in this paper is very good and it highlights that the protein microarray can be used to rapidly generate data relevant to the COVID-19 outbreak, and other outbreaks in the future. But the interpretation and generalizations in this draft are misleading and may cause future confusion. The amino acid sequences and their exact position in the genome need to be clearly labeled.

Reviewer #2:

Remarks to the Author:

In their manuscript Jiang and colleagues constructed a protein microarray that allows to interrogate the immune response to the majority of SARS-CoV-2 proteins. The work is very interesting and should be published asap. However, there are many points that need the authors' attention.

Major points

1) The language needs editing by a professional editing service. In addition, many abbreviations are not defined.

2) The data are nice but the figures are chaotic. The following should be changed:

a) The labelling of the proteins throughout needs to be more systematic. It needs to be clear what is what, the codes used are not helpful. The expression system used needs to be indicated as well.

b) In general, it would be good if the graphs in the different panels would have labels so that the reader understands what she/he is looking at.

c) Figures 3 and 4 need to be restructured. To allow comparison of reactivity to different

constructs for each protein, the reactivity should be clustered by protein.

d) eGFP has signal in some samples/rows. Is this a contamination?

3) The negative controls need to be shown in Figures 3 and 4 as well.

Minor points

1) Line 51 and 52: The numbers should be updated.

2) The S protein should only count as one protein. Also, recombinant full length S protein should be included in the analysis.

3) Line 67: Remdesivir has gotten EUA by the FDA.

4) Lines 79, 80 etc.: Is it really 'nucleocapsid'? Isn't it really a nucleoprotein?

5) Line 95: This has also been used nicely for influenza virus. A good reference would be <https://www.ncbi.nlm.nih.gov/pubmed/31964741>

6) Line 112: The headline is unclear.

7) Line 163: Please separate the two words here.

8) Line 320 and 342: Better specificity with mammalian versus insect cell expressed proteins and better signal with full length S than with RBD was also recently demonstrated by Amanat et al.: <https://www.medrxiv.org/content/10.1101/2020.03.17.20037713v2>

9) Figure 1 could certainly be a supplementary figure.

Reviewer #3:

Remarks to the Author:

The manuscript by He-wei Jiang et al. describes a study in which 18 of the SARS-CoV-2 proteins were analyzed (38 proteins in total because some proteins were added multiple times, when they were obtained from different sources or when parts were expressed). Most were expressed as his- or GST-tagged recombinant proteins, followed by quality control and printing onto a protein microarray. Convalescent sera from 29 COVID-19 patients and 21 controls were analysed to profile the IgG and IgM responses to these proteins. All COVID-19 patients showed responses to especially N and S1.

The manuscript addresses a currently very relevant issue regarding immunological responses

to SARS-CoV-2 infection and their correlation with the outcome of infection, which is of interest to a broad audience involved in SARS-CoV2 research, diagnostics and patient care.

Language could be improved and the manuscript requires editing by a native speaker before it could be published (do not forget the titles of the Results section). For example, it is not easy to understand what the authors mean with sentences like line 215-217.

A strong point of the study is that the quality of the arrays and reproducibility is first assessed (line 142-147).

Overall, the IgG and IgM responses to S1, S-RBD and N that are observed for the COVID-19 patients look convincing. However, there are some concerns about the data analysis, especially regarding the number and type of statistical tests that were performed. Consequently, the claims that are made about correlations of the antibody response with age, LDH and lymphocyte percentage are not that convincing (yet).

Specific issues

In the statistical analysis there is no correction performed for multiple testing which is standard practice in the microarray field. In this manuscript many different tests are performed so a correction is really essential to be able to determine whether the differences or correlations that are found are truly relevant. From the materials and methods section it is also unclear which type of statistical test was used to test differential antibody response. A list with the test results for all proteins would be very useful.

In Figure 7 and Figure S4 Pearson correlations are used to determine relationships between certain clinical characteristics and the IgG response. A regression model would be more suitable for this purpose. A regression model can also be used to test the influence of certain confounding factors such as age or sex on the outcome instead of using only a data subset in an analysis (as was e.g. done with only males or females, or a subset of the females). The amount of tests that are performed using subsets of already quite small numbers of patients poses the question whether the reported correlations are truly significant/relevant.

Including a table listing all the proteins that were used, in particular which variants were included would help the reader to more easily get the overview of what was tested. E.g. nsp7 is shown in the gels of Figure S1 and the array in Figure 2, but in the heatmap it isn't present. From the methods section (Line 309) it appears that this protein was removed from the analysis due to technical issues. So was this one of the 18 proteins that was analysed according to the text, or are there actually only 17 for which data was obtained?

Please explain the choice for the control sera, why are healthy controls and lung cancer patients combined? The lung cancer patients do not seem to be a very good match in age with the COVID-19 patients. These control sera were obtained from another hospital, were they

collected and treated in the same manner? Were they collected in the same period or obtained from a biobank?

Based on what do the authors conclude (e.g. in line 173) that the IgG response is much stronger? IgG and IgM are detected with different secondary antibodies which might have different affinities and are also coupled to different fluorophores. Was some kind of calibrator (standard curve) used? The IgM and IgG signal cannot be compared (unlike e.g. all IgM signals of different sera).

Based on Fig S1, the purity of several proteins appears to be rather low (e.g. nsp8 and nsp10). This raises questions regarding the specificity of some of the signals. Were negative control 'lysates' (purification of empty vector or GFP) included as controls? Western blots with some patient sera could also be done to show that the sera recognize proteins of the correct size.

Line 27: a causal relationship between IgG & IgM response and outcome of disease (correlates of protection) has not been proven, so authors cannot claim that "patients were recovered due to their IgG and IgM" response.

Line 83: Please remove "And, etc." When there are more questions that are important for the reader to know about please write these out and do not leave them up to the imagination.

Line 121/122: It would be helpful to the reader to list somewhere what the expected sizes of the proteins is.

Line 157: What were the standard criteria? Please add a reference or briefly describe.

Line 165 and 204/205: Please indicate in the text which other proteins also elicited responses.

Line 185: Based on what criteria did the authors conclude that "proteins were all good performance for detection"?

Line 206: Should under be above? How were these thresholds determined? Please add this to the methods section, it can be found in the figure legend, but that is not sufficient.

Line 123 and Line 380-382: Line 380-382 is confusing because according to line 123 and table S1 several of the S proteins in the array were already produced in a mammalian system so these samples are already expected to be glycosylated. Was the presence of N-glycosylation sites on these proteins tested? E.g. through digestion with PNGaseF?

Figure 2B: It would be useful to also have the layout of the microarray indicated for the COVID-19 patient and normal control arrays. This would make it easier for the reader to see which of the green dots correspond to which protein. This could be done by breaking the array up as is done in Figure 2A or these examples can be combined with Figure 2A by showing the corresponding parts of the arrays that are now Figure 2B below the array parts of Figure 2A.

In many of the figures differently coloured dots (red, blue and green) are used to indicate that data points belong to different data sets. The use of different shapes instead of different colours, e.g. dots, triangles and squares, might help (colour blind) readers to more easily distinguish which data point belongs to which data set, in particular when the paper is printed in black and white.

Reviewer #1 (Remarks to the Author):

This is a good paper showing that protein microarrays can be applied to rapidly generate outbreak data relevant to the development of diagnostic test, serosurveillance, and vaccine discovery.

Figure 3 has much of the data from this study in a heat map. Most of the antigen features on the array are either non-structural or the N protein or S1. The antigen feature labels are not clear enough and a reviewer cannot translate what each one is. The proteins should all be given in a table or spreadsheet with their amino acid coordinates.

Response : *We thank this reviewer for the appreciation of our work. Indeed, **Fig. 3** was somewhat complicate with lot of detailed information. To clarify, we have now simplified and rearranged **Fig. 3** and **Fig. 4**. The previous version of the two heatmaps are rich of information, which may facilitate the understanding of our findings at a global level, thus, we decided to keep them, but as supplementary Figures (**Supplementary Fig. 3 and 4**). The detailed information of the proteins are shown in **Supplementary Table 1**, including nucleotide sequences, amino acid sequences, the expected sizes of the proteins and etc. We have now emphasized this information in the revision (**Marked Version line 128**).*

All of the N protein variants discriminate between COVID cases and non-COVID controls. The non-structural proteins are not immunogenic and they do not recognized antibodies that differentiate cases and controls. It is not clear from the antigen labels what each feature is. The authors conclude that S2 is not reactive, but the heatmap seems to show that it is differentially reactive.

Response : *We thank the reviewer for pointing this out. As shown in **Fig. 6a**, some of the non-structural proteins, for example NSP5 and NSP14 were immunogenic in some of the patients. However, they may not be good candidates for diagnostic purpose. The labels was not that clear, to clarify, we have now added a layout table to **Fig. 2a**. and rearranged **Fig. 2**. For S2, I guess what the reviewer referred is the description “the S1 signals were poorly correlated with S2 proteins” on previous line 192. Actually, significant IgG/IgM responses were observed for S2, though not for 100% of the patients. We have now modified the description as “the S1 signals were poorly correlated with S2 proteins, though significant S2 signals were observed for many of the patients”. (**Marked Version line 210-212**).*

The data in this paper is very good and it highlight that the protein microarray can be used to rapidly generate data relevant to the COVID-19 outbreak, and other outbreaks in the future. But the interpretation and generalizations in this draft are misleading and may cause future confusion. The amino acid sequences and their exact position in the

genome need to be clearly labeled.

Response : *We thank the reviewer for the appreciation of the protein microarray. We guess what the reviewer referred is **Fig. 1a**, the purpose of this panel is to show the relative locations and sizes of the proteins encoded by the SARS-CoV-2 genome. We would like to keep it as is. However, we agree with this reviewer that the position information of the protein coding gene will be useful, thus we have now added this information to **Supplementary Table 1**, which already contains the amino acid sequences of all the proteins.*

Reviewer #2 (Remarks to the Author):

In their manuscript Jiang and colleagues constructed a protein microarray that allows to interrogate the immune response to the majority of SARS-CoV-2 proteins. The work is very interesting and should be published asap. However, there are many points that need the authors' attention.

Major points

1) The language needs editing by a professional editing service. In addition, many abbreviations are not defined.

Response : *We thank this reviewer for the appreciation of our work. We added some abbreviations definition to the revised manuscript and invited a professional of native English speaking to edit our manuscript.*

2) The data are nice but the figures are chaotic. The following should be changed:

a) The labelling of the proteins throughout needs to be more systematic. It needs to be clear what is what, the codes used are not helpful. The expression system used needs to be indicated as well.

Response : *We have checked the labeling of proteins throughout the manuscript and made appropriate modifications to make them consistent. We guess what “codes” that the reviewer referred to is “**T**: Tao Lab; **B**: Hangzhou Bioeast biotech. Co.,Ltd.; **K**: Healthcode Co., Ltd.; **S**: Sanyou biopharmaceuticals Co.,Ltd.; **W**: VACURE 1 Biotechnology Co.,Ltd. **Y**: Sino biological Co.,Ltd.” . We think it will be too busy to keep these information directly in the Figures, thus, we would like to keep these “codes”. However, to facilitate the understanding of our work, we have now added the annotations of these codes “**T**: Tao Lab; **B**: Hangzhou Bioeast biotech. Co.,Ltd.; **K**: Healthcode Co.,Ltd.; **S**: Sanyou biopharmaceuticals Co.,Ltd.; **W**: VACURE 1 Biotechnology Co.,Ltd. **Y**: Sino biological Co.,Ltd.” to the figure legend of **Fig. 2**.*

*And also the expression system. For more detailed information, the reviewer may check **Supplementary Table 1**.*

b) In general, it would be good if the graphs in the different panels would have labels so that the reader understands what she/he is looking at.

Response : *We thank the reviewer for this suggestion. We have now added appropriate labels to many of the panels.*

c) Figures 3 and 4 need to be restructured. To allow comparison of reactivity to different constructs for each protein, the reactivity should be clustered by protein.

Response : *Indeed, **Fig. 3** and **Fig. 4** was somewhat complicate with lot of detailed information. To clarify, we have now simplified and rearranged **Fig. 3** and **Fig. 4**. The previous version of the two heatmaps are rich of information, which may facilitate the understanding of our findings at a global level, thus, we decided to keep them, but as supplementary Figures (**Supplementary Fig. 3** and **4**). More detailed comparisons of reactivity to different constructs for each protein were shown in **Fig. 5**, **Supplementary Fig. 5** and **6**.*

d) eGFP has signal in some samples/rows. Is this a contamination?

Response :

*The eGFP at 0.5 mg/ml printing concentration has some moderate signals in some samples, while the same protein with 0.25 and 0.1 mg/ml do not. It might be from contamination. Actually, in the serum incubation buffer, eGFP and E. coli lysates were added to eliminate the otherwise observed unspecific signals for eGFP tagged proteins. We added a **supplementary Figure 2** to demonstrate this point in the revised manuscript. In addition, for other proteins with eGFP tag, the spots with different concentrations have consistently signals in the samples and the proteins at 0.5 mg/ml are occasionally positive in the samples which have moderate signals for eGFP at 0.5 mg/ml. These observations demonstrate the signals for the proteins with eGFP tag are majorly due to the target proteins, but not eGFP tag.*

3) The negative controls need to be shown in Figures 3 and 4 as well.

Response : *We are sorry for confusion. Indeed, there are some negative controls, such as eGFP and GST. To improve the clarity, we have now simplified **Fig. 3** and **Fig. 4**.*

Minor points

1) Line 51 and 52: The numbers should be updated.

Response : *We have updated the numbers.*

2) The S protein should only count as one protein. Also, recombinant full length S protein should be included in the analysis.

Response : *According to Fig. 1a , S protein was indeed counted as one protein. We have now checked throughout the manuscript and make it consistent. We agree with the reviewer that full length S protein should be included, actually, we have tried to purify the full length S protein, however, we failed even after several rounds of optimization.*

3) Line 67: Remdesivir has gotten EUA by the FDA.

Response : *Remdesivir is a promising drug candidate for treating COVID-19. There are several studies about its effectiveness, however, the conclusions are still controversial.*

Refs and links:

1.<https://www.gilead.com/news-and-press/press-room/press-releases/2020/4/gilead-announces-results-from-phase-3-trial-of-investigational-antiviral-remdesivir-in-patients-with-severe-covid-19>

2.[https://www.thelancet.com/journals/lancet/article/PIIS0140-6736\(20\)31022-9/fulltext](https://www.thelancet.com/journals/lancet/article/PIIS0140-6736(20)31022-9/fulltext)

3.<https://www.niaid.nih.gov/news-events/nih-clinical-trial-shows-remdesivir-accelerates-recovery-advanced-covid-19>

4.<https://www.nature.com/articles/d41586-020-01295-8>

4) Lines 79, 80 etc.: Is it really nucleocapsid? Is it really a nucleoprotein?

Response : *Yes, N protein is a nucleocapsid protein.*

5) Line 95: This has also been used nicely for influenza virus. A good reference would be <https://www.ncbi.nlm.nih.gov/pubmed/31964741>

Response : *We thank the reviewer for the recommendation. We have now cited this reference. (Marked Version line 92)*

6) Line 112: The headline is unclear.

Response : *We thank the reviewer for the recommendation. We revised this headline.*

7) Line 163: Please separate the two words here.

Response : *We revised this.*

8) Line 320 and 342: Better specificity with mammalian versus insect cell expressed proteins and better signal with full length S than with RBD was also recently demonstrated by Amanat et al.: <https://www.medrxiv.org/content/10.1101/2020.03.17.20037713v2>

Response : *We thank the reviewer for the recommendation, we have now cited this reference. (Marked Version line 390 and line 414)*

9) Figure 1 could certainly be a supplementary figure.

Response : *Since Fig. 1 shows the overall design and workflow of this study, we prefer to keep it as is.*

Reviewer #3 (Remarks to the Author):

The manuscript by He-wei Jiang et al. describes a study in which 18 of the SARS-CoV-2 proteins were analyzed (38 proteins in total because some proteins were added multiple times, when they were obtained from different sources or when parts were expressed). Most were expressed as his- or GST-tagged recombinant proteins, followed by quality control and printing onto a protein microarray. Convalescent sera from 29 COVID-19 patients and 21 controls were analysed to profile the IgG and IgM responses to these proteins. All COVID-19 patients showed responses to especially N and S1.

The manuscript addresses a currently very relevant issue regarding immunological responses to SARS-CoV-2 infection and their correlation with the outcome of infection, which is of interest to a broad audience involved in SARS-CoV2 research, diagnostics and patient care.

Language could be improved and the manuscript requires editing by a native speaker before it could be published (do not forget the titles of the Results section). For example, it is not easy to understand what the authors mean with sentences like line 215-217.

A strong point of the study is that the quality of the arrays and reproducibility is first assessed (line 142-147).

Overall, the IgG and IgM responses to S1, S-RBD and N that are observed for the COVID-19 patients look convincing. However, there are some concerns about the data analysis, especially regarding the number and type of statistical tests that were

performed. Consequently, the claims that are made about correlations of the antibody response with age, LDH and lymphocyte percentage are not that convincing (yet).

Response: *We thank the reviewer for the positive evaluation of our work.*

Specific issues

In the statistical analysis there is no correction performed for multiple testing which is standard practice in the microarray field. In this manuscript many different tests are performed so a correction is really essential to be able to determine whether the differences or correlations that are found are truly relevant. From the materials and methods section it is also unclear which type of statistical test was used to test differential antibody response. A list with the test results for all proteins would be very useful.

Response :*We agree with the reviewer it is a common practice to perform correlation/normalization for microarray data as what we and others have done before (Yang Li, et al., EbioMedicine, 2020;53:102674; Jianbo Pan, et al., MCP, 2017;16: 2069–2078). However, we haven't perform any normalization between arrays in the present study. The reasons are as follows. Firstly, all the data that we presented were generated from the same batch of microarrays, all the sera were analyzed in a single experiment. The initial evaluation showed that microarray data were highly consistent. Secondly, to judge the reproducibility , two positive sera and blank controls were also analyzed in triplicate on the microarray. The data showed that the variation among these repeats was negligible.*

In Figure 7 and Figure S4 Pearson correlations are used to determine relationships between certain clinical characteristics and the IgG response. A regression model would be more suitable for this purpose. A regression model can also be used to test the influence of certain confounding factors such as age or sex on the outcome instead of using only a data subset in an analysis (as was e.g. done with only males or females, or a subset of the females). The amount of tests that are performed using subsets of already quite small numbers of patients poses the question whether the reported correlations are truly significant/relevant.

Response: *We appreciate the reviewer's constructive suggestion. We have now used multiple linear regression to determine the relationships between the IgG response and the clinical characteristics (i.e., age, gender, days after onset, peak LDH and Ly%). Consistent with our correlation analysis, age and peak LDH are statistically significant (both with p-values <0.05) and gender shows marginally significance (p = 0.059). Days after onset, which has been identified as a confounding factor, shows*

no statistical significance ($p = 0.514$) and thus, it is removed from the regression. Ly% is still kept in the model as its relatively low significance ($p = 0.106$) is probably due to the small sample size. The regression model has an adjusted R-squared of 0.6 and a p-value < 0.001 . Overall, the results of multiple regression are in accordance with our previous results. We have added the regression analysis into the results and methods of the revised manuscript (**Marked Version line 278-288 and line 376-381**).

Including a table listing all the proteins that were used, in particular which variants were included would help the reader to more easily get the overview of what was tested. E.g. nsp7 is shown in the gels of Supplementary Fig. 1 and the array in Fig. 2, but in the heatmap it is not present. From the methods section (Line 309) it appears that this protein was removed from the analysis due to technical issues. So was this one of the 18 proteins that was analysed according to the text, or are there actually only 17 for which data was obtained?

Response : *The detailed information of the proteins on the microarray are included in **Supplementary Table 1**. The reviewer is right, nsp7 was purified and printed on the microarray. However, after we finished the probing of the sera, according to our experience, we noticed that nsp7 was contaminated during the microarray manufacturing process. Thus we decided not to include nsp7 for further analysis, and the reviewer is right, there are actually 17 for which data was obtained. To clarify, we have modified the manuscript wherever necessary (**Marked Version line 148-150**). This problem has already been resolved in the new version of the SARS-CoV-2 protein microarray.*

Please explain the choice for the control sera, why are healthy controls and lung cancer patients combined? The lung cancer patients do not seem to be a very good match in age with the COVID-19 patients. These control sera were obtained from another hospital, were they collected and treated in the same manner? Were they collected in the same period or obtained from a biobank?

Response : *These sera were collected before the outbreak of COVID-19 at Shanghai. (**Table 1**) They were collected in the same manner during 7/2018-8/2019 and initially for another project. Lung cancer patients were chosen with the consideration that like COVID-19, it is also mainly associated with lung. According to the results, there was no statistical difference between lung cancer patients and healthy controls in terms of antibody response against proteins of SARS-CoV-2, so they may could be combined as a negative control group. The average age of the lung cancer group is older than the COVID-19 and healthy control group. For the same reason as mentioned above, we think this might not affect our main conclusions.*

Based on what do the authors conclude (e.g. in line 173) that the IgG response is

much stronger? IgG and IgM are detected with different secondary antibodies which might have different affinities and are also coupled to different fluorophores. Was some kind of calibrator (standard curve) used? The IgM and IgG signal cannot be compared (unlike e.g. all IgM signals of different sera).

Response : *Because only convalescent sera were analyzed in this study, it is known that IgM level is generally lower than that of IgG at the later stage of infection, we think it is reasonable to argue we could observe the same phenomena for our microarray analysis. However, we agree with the reviewer, because the fluorescent secondary antibodies and the scanning settings are different for IgG and IgM, without an accurate calibrator, it is hard to perform quantitative comparison between IgG and IgM. We have now removed this sentence.*

Based on Fig S1, the purity of several proteins appears to be rather low (e.g. nsp8 and nsp10). This raises questions regarding the specificity of some of the signals. Were negative control 'lysates' (purification of empty vector or GFP) included as controls? Western blots with some patient sera could also be done to show that the sera recognize proteins of the correct size.

Response : *We thank the reviewer for this critical comment. Because many of the proteins were fused with eGFP or GST, to rule out the possibility that these two tags may cause non-specific signals, eGFP and GST were also included in the microarray as negative controls. In addition, to minimize the possible nonspecific background that may be caused by eGFP or co-purified E. coli proteins, E.coli lysate and eGFP were included in the incubation buffer when probing sera on the microarray. Indeed, some of the nonspecific signals disappeared upon the addition of E.coli lysate and eGFP. We have now added this result as **Supplementary Fig. 2**. The details of the addition of E. coli lysate and eGFP in the incubation buffer were also added to the methods section (**Marked Version line 355-357**)*

Line 27: a causal relationship between IgG & IgM response and outcome of disease (correlates of protection) has not been proven, so authors cannot claim that "patients were recovered due to their IgG and IgM" response.

Response : *We thank the reviewer for this comment. We agree with this reviewer, we have revised this description (**Marked Version line 27-28**).*

Line 83: Please remove And, etc. When there are more questions that are important for the reader to know about please write these out and do not leave them up to the

imagination.

Response: *We agree with this reviewer, we have revised this description.*

Line 121/122: It would be helpful to the reader to list somewhere what the expected sizes of the proteins is.

Response : *We agree with this reviewer, please see **Supplementary Table 1** for the detailed information of these proteins.*

Line 157: What were the standard criteria? Please add a reference or briefly describe.

Response : *The standard criteria is **Diagnosis and Treatment Protocol for Novel Coronavirus Pneumonia (Trial Version 5)**, released by the National Health Commission & State Administration of Traditional Chinese Medicine. Serum from each patient was collected on the day of hospital discharge when the standard criteria were met. We have added the description of the criteria to methods (**Marked Version line 337-347**).*

Briefly, the key points for the discharge criteria are:

- 1) Body temperature is back to normal for more than three days;*
- 2) Respiratory symptoms improve obviously;*
- 3) Pulmonary imaging shows obvious absorption of inflammation,*
- 4) Nuclei acid tests negative twice consecutively on respiratory tract samples such as sputum and nasopharyngeal swabs (sampling interval being at least 24 hours).*

Line 165 and 204/205: Please indicate in the text which other proteins also elicited responses.

Response : *We agree with this reviewer, we have revised this description. (**Marked Version line 181 and line 225**)*

Line 185: Based on what criteria did the authors conclude that "proteins were all good performance for detection"?

Response: *We thank the reviewer for pointing this out. Because we found that for full-length S1 proteins from different sources, i.e., from **E.coli** (S1_T) or **293T** (S1_B and S1_S) expression system, high correlations with each other were observed. Thus we conclude that all these S1 proteins were good for detection. To clarify, we have now modified the manuscript. (**Marked Version line 200-204**)*

Line 206: Should under be above? How were these thresholds determined? Please add this to the methods section, it can be found in the figure legend, but that is not sufficient.

Response: *We thank the reviewer for pointing this out. Should be “above”. To calculate the positive rate of antibody response for each protein, mean signal + 3*SD of the control sera were used to set the threshold. We have refined the description and added it to the method section ((Marked Version line 376-381).*

Line 123 and Line 380-382: Line 380-382 is confusing because according to line 123 and table S1 several of the S proteins in the array were already produced in a mammalian system so these samples are already expected to be glycosylated. Was the presence of N-glycosylation sites on these proteins tested? E.g. through digestion with PNGaseF?

Response : *We are sorry for the confusion. Only a few proteins on the current protein microarray were prepared using mammalian cell systems. We are trying the rest of the proteins. We have not tested the glycosylation of these proteins. To clarify, we have modified the descriptions ((Marked Version line 454-458).*

Figure 2B: It would be useful to also have the layout of the microarray indicated for the COVID-19 patient and normal control arrays. This would make it easier for the reader to see which of the green dots correspond to which protein. This could be done by breaking the array up as is done in Figure 2A or these examples can be combined with Figure 2A by showing the corresponding parts of the arrays that are now Figure 2B below the array parts of Figure 2A.

Response : *We agree with this reviewer and revised Fig. 2.*

In many of the figures differently coloured dots (red, blue and green) are used to indicate that data points belong to different data sets. The use of different shapes instead of different colours, e.g. dots, triangles and squares, might help (colour blind) readers to more easily distinguish which data point belongs to which data set, in particular when the paper is printed in black and white.

Response : *We thank this reviewer for this considerate suggestion. We have modified the Figures wherever necessary that may cause difficulties for (colour blind) readers. Like Fig. 5d, h, 6d, e, 7b and etc.*

Reviewers' Comments:

Reviewer #1:

Remarks to the Author:

The authors responded in detail to all of the reviewer's comments.

Reviewer #2:

Remarks to the Author:

The authors did a good job in addressing the reviewers comments. Some editing including in the figures might still be needed.

Reviewer #3:

Remarks to the Author:

The manuscript by Jiang et al. has been improved by the authors, it has become easier to read and they have satisfactorily addressed most of the issues that were raised.

The issue regarding the specificity of the signal on the microarray has been addressed, by supplying important information on the method (addition of E coli lysate and GFP in incubation buffer) that was lacking in the original manuscript and by providing supplemental figure 2. A supplemental figure with a simple immunoblot, similar to one already present in the paper with all proteins, probed with one of the patient sera to show that proteins of the correct size are recognized could be considered to strengthen claims on specificity.

The question regarding multiple testing was not answered (a correction for multiple testing is not the same as a normalization step). I wonder whether the mean signal + 3*SD of the control sera threshold is the best way to analyse this dataset. For example, in figure 5A the Log₂(FI) does not look normally distributed for all of the proteins in the control samples. This does not mean that the findings are incorrect; there are simply several methods available that were specifically developed for the analysis of microarray data, e.g. significance analysis of microarrays (SAM), which are probably better suited to do the analysis.

The authors have now added a multiple linear regression model analysis to the paper which is good as this is a much better way to test the contribution of clinical parameters to an outcome. This analysis should be shown instead of the subset analyses that they did before. It should not be an addition to their previous analyses. The Pearson correlations in Figure 7 and S7 should be removed.

Reviewer #3

The manuscript by Jiang et al. has been improved by the authors, it has become easier to read and they have satisfactorily addressed most of the issues that were raised.

The issue regarding the specificity of the signal on the microarray has been addressed, by supplying important information on the method (addition of E coli lysate and GFP in incubation buffer) that was lacking in the original manuscript and by providing supplemental figure 2. A supplemental figure with a simple immunoblot, similar to one already present in the paper with all proteins, probed with one of the patient sera to show that proteins of the correct size are recognized could be considered to strengthen claims on specificity.

Response:

We thank this reviewer for this constructive suggestion. To further prove the specificity, we performed the immunoblot as suggested by the reviewer. Briefly, we selected several proteins which showed positive signals on the microarray, e.g., S protein, N protein and ORF9b to prepare a SDS-PAGE gel, several proteins which showed negative signal on the microarray were also selected, but as negative controls. A serum sample which showed positive binding on the microarray was included to make the immunoblot. According to our microarray results, the signals of S protein and N protein were usually much higher than that of other proteins, such as ORF9b. In order to avoid the subsequent signal oversaturation, we included the S and N proteins in one blot and the rest in another blot. As expected, the serum specifically recognized

ORF9b, S proteins and N proteins. Interestingly, S-RBD exhibited a relatively lower immunoblotting signal compared to that of S1 proteins. (please see **Supplementary Fig. 2b**).

*The question regarding multiple testing was not answered (a correction for multiple testing is not the same as a normalization step). I wonder whether the mean signal + 3*SD of the control sera threshold is the best way to analyse this dataset. For example, in figure 5A the Log₂(F_i) does not look normally distributed for all of the proteins in the control samples. This does not mean that the findings are incorrect; there are simply several methods available that were specifically developed for the analysis of microarray data, e.g. significance analysis of microarrays (SAM), which are probably better suited to do the analysis.*

Response:

We thank the reviewer for this constructive suggestion. We have now applied BH (Benjamini-Hochberg) method for multiple testing and acquire adjusted p value or q value for all the proteins. We have also performed SAM analysis to identify significant positive proteins. A list of the proteins with adjusted p value and SAM results was added (**Supplementary Table 2**) in the revised manuscript. The results demonstrated that all the analysis were highly consistent among p value, q value and the score of SAM. SAM plot was added as well (**Supplementary Fig. 7**). Accordingly, the main text and figures were modified wherever necessary. The mean signal + 3*SD of the control sera threshold may not be the best way to analyze our data. However, it is very common in the protein microarray field (or microarray field) to call hits by using “mean signal + 3*SD of the control” as the cut off.

The authors have now added a multiple linear regression model analysis to the paper which is good as this is a much better way to test the contribution of clinical parameters to an outcome. This analysis should be shown instead of the subset analyses that they did before. It should not be an addition to their previous analyses. The Pearson correlations in Figure 7 and S7 should be removed.

Response:

We thank the reviewer for this suggestion. We have now included the multiple linear regression model in **Fig. 7**. According to the reviewer’s suggestion, most of the correlation

analysis results of the subsets were removed from **Fig. 7**. We think some of the correlation analysis may still provide meaningful information, we would like to keep them in **Fig. 7**. To focus on the major points, the previous supplementary Fig. 7 was also removed.

Reviewers' Comments:

Reviewer #3:

Remarks to the Author:

The authors have performed the suggested additional experiment and have gone a long way to address all previous issues that were raised, including revision of the statistical analysis. They have satisfactorily addressed all the issues that were raised.